

# Preliminary analysis of New Zealand scampi (*Metanephrops challengeri*) diet using metabarcoding

Aimee L. van der Reis[1], Olivier Laroche[2], Andrew G. Jeffs[1,2] and Shane D. Lavery[1,2]

[1] Institute of Marine Science, University of Auckland, Auckland, New Zealand
[2] School of Biological Sciences, University of Auckland, Auckland, New Zealand

## ABSTRACT

Deep sea lobsters are highly valued for seafood and provide the basis of important commercial fisheries in many parts of the world. Despite their economic significance, relatively little is known about their natural diets. Microscopic analyses of foregut content in some species have suffered from low taxonomic resolution, with many of the dietary items difficult to reliably identify as their tissue is easily digested. DNA metabarcoding has the potential to provide greater taxonomic resolution of the diet of the New Zealand scampi (*Metanephrops challengeri*) through the identification of gut contents, but a number of methodological concerns need to be overcome first to ensure optimum DNA metabarcoding results. In this study, a range of methodological parameters were tested to determine the optimum protocols for DNA metabarcoding, and provide a first view of *M. challengeri* diet. Several PCR protocols were tested, using two universal primer pairs targeting the 18S rRNA and COI genes, on DNA extracted from both frozen and ethanol preserved samples for both foregut and hindgut digesta. The selection of appropriate DNA polymerases, buffers and methods for reducing PCR inhibitors (including the use of BSA) were found to be critical. Amplification from frozen or ethanol preserved gut contents appeared similarly dependable. The COI gene was found to be more effective than 18S rRNA gene for identifying large eukaryotic taxa from the digesta; however, it was less successfully amplified. The 18S rRNA gene was more easily amplified, but identified mostly smaller marine organisms such as plankton and parasites. This preliminary analysis of the diet of *M. challengeri* identified a range of species (13,541 reads identified as diet), which included the ghost shark (*Hydrolagus novaezealandiae*), silver warehou (*Seriolella punctata*), tall sea pen (*Funiculina quadrangularis*) and the salp (*Ihlea racovitzai*), suggesting that they have a varied diet, with a high reliance on scavenging a diverse range of pelagic and benthic species from the seafloor.

Corresponding author
Aimee L. van der Reis,
avan398@aucklanduni.ac.nz

## INTRODUCTION

Commercial fisheries for deep sea lobster species, those typically captured below 50 m depth, are occurring in many parts of the world, currently producing total annual landings of around 50,000 t (*Jeffs, 2010*; *FAO, 2016*). Many of the targeted species are of high value, with wholesale export prices of over \$40 kg$^{-1}$ common for some species, such as the New Zealand (NZ) scampi *Metanephrops challengeri* (*Balss, 1914*; *Seafood New Zealand, 2017*). Despite the economic importance of deep sea lobsters, the knowledge of their feeding ecology and diet is limited. Improved information on diet in deep sea lobsters has potential uses in identifying effective baits for trap fishing, development of suitable feeds for aquaculture, as well as understanding differences in growth rates in natural populations to help improve the management of the fisheries.

Species in two genera of deep-sea lobster, *Metanephrops* and *Nephrops*, are widely targeted by commercial fisheries, share similar ecology and are genetically closely related (*Tshudy, Chan & Sorhannus, 2007*). The Norway lobster, *Nephrops norvegicus,* has a varied diet, which is necessary to achieve optimal growth (*Cristo & Cartes, 1998*; *Mente, 2010*). Similar varied diets have been suggested for some *Metanephrops* species (excluding *M. challengeri*), and include fish, crustaceans, polychaetes and amphipods, although these have only been identified at a crude taxonomic level using microscopic analysis of foregut contents (*Choi et al., 2008*; *Sahlmann, Chan & Chan, 2011*; *Wahle et al., 2012*; *Bell, Tuck & Dobby, 2013*). A significant problem with microscopic analysis of gut contents is poor taxonomic resolution, due to difficulties in identification of partly-digested specimens, which typically require an expert for reliable taxonomic identification (*Dunn et al., 2010*; *Pompanon et al., 2012*; *Zhan et al., 2013*; *Berry et al., 2015*; *Young et al., 2015*; *Crisol-Martínez et al., 2016*; *Harms-Tuohy, Schizas & Appeldoorn, 2016*; *Sousa et al., 2016*). Also, soft-bodied animals are frequently suspected of being under-represented in such analyses as they are highly digestible (*Bell, Tuck & Dobby, 2013*).

A more promising tool for diet analysis, DNA metabarcoding, combines universal DNA primers with high-throughput (next-generation) sequencing to identify a variety of species from a mixture of gut content DNA (*Kress et al., 2015*). A single universal primer pair has the ability to amplify a diverse range of species by targeting a single gene region that has been conserved among phylogenetically distinct taxa (e.g., the cytochrome oxidase I [COI] region in the mitochondrial DNA or the 18S ribosomal RNA [rRNA] region in the nuclear DNA; *Aylagas et al., 2016*). This molecular method has been successfully applied in diet studies of a variety of lobster larvae and marine fish species (*O'Rorke et al., 2012*; *O'Rorke et al., 2014*; *Berry et al., 2015*; *Harms-Tuohy, Schizas & Appeldoorn, 2016*). One of the major advantages of this method of gut analysis is that a high degree of taxonomic resolution can be achieved, with the identification of individual species frequently possible. Also, the digested or liquid gut content, which would normally be of no value for visual identification of morphology, can provide additional dietary information that would otherwise have been missed. The potential difficulties of this approach include the sequencing and labour costs, the sensitivity of detection, and the taxonomic coverage of the reference sequence databases

at the time of study. These difficulties will likely fade as research progresses in this field (*Cowart et al., 2015*; *Srivathsan et al., 2016*).

With sequencing cost and sensitivity being the major restrictions, it is in the best interest of researchers to first address methodological issues before a full analysis is conducted, in order to ensure the best possible results are obtained. One of the potential difficulties in metabarcoding is polymerase chain reaction (PCR) inhibition, which has often been found when using template genomic DNA extracted from biological material with a high proportion of organic (i.e., bile salts) or inorganic (i.e., calcium ions) compounds, or from body fluids or some difficult organs (*Rossen et al., 1992*; *Kreader, 1996*; *Rådström et al., 2004*; *Farell & Alexandre, 2012*; *Schrader et al., 2012*). Another matter to address is determining which DNA polymerases are more susceptible to the effects of specific PCR inhibitors present that impede DNA amplification (*Rådström et al., 2004*). A third potential problem is the taxonomic coverage and resolution possible from the different reference sequences in the databases available, with some databases targeting different genes or different organisms. Other sampling factors may also be important in determining the success of metabarcoding for diet analysis. These include the preservation method (e.g., frozen or in ethanol) and the location of the gut contents to be analysed (from the foregut or hindgut), with the potential for these two locations to provide different results due either to the state of DNA degradation or the differences in timing of intake and digestive processing of dietary items.

The natural diet of *M. challengeri* is largely unknown, despite forming the basis of an important commercial fishery in New Zealand, and currently being of some interest for aquaculture development. The goal of this study is to provide a preliminary examination of the natural diet of this species using metabarcoding methods, by firstly addressing the optimisation of selected methodological factors, including issues of tissue choice and preservation, target gene, PCR inhibition, PCR reagents and reference database coverage.

## MATERIALS & METHODS

### Sample collection and locations

Samples of *M. challengeri* were collected in September of 2016 from the seafloor of the Chatham Rise, New Zealand, at depths ranging from 200–500 m from the R.V. Kaharoa by towing benthic trawl nets with a 45 mm cod-end. Upon landing the trawl on the deck, the *M. challengeri* that were intact were euthanized by immersing them in chilled 95% ethanol as soon as possible to avoid any degradation. The samples were transferred to the University of Auckland's laboratories (Auckland, New Zealand) for further examination. A single trawl location (start −42.9952°, 177.2367° at a depth of 322 m; end −43.0210°, 177.1800° at a depth of 252 m) and the individuals collected there were selected for the purpose of this study. For comparison, frozen individuals of commercially harvested *M. challengeri* were supplied by Sanford Ltd (Auckland, New Zealand) that were collected from the seafloor using a benthic trawl on the continental shelf off New Zealand's Auckland Islands (the trawl took place within a quadrant: corner A −49.0°, 168.5°; corner B −51.5°, 168.5°; corner C −52.0°, 164.5°; corner D −49.0°, 164.5°) and snap frozen immediately after collection.

A special permit (#549) for *M. challengeri* collection was provided by New Zealand's Ministry of Primary Industries. The specimens for this study were collected in accordance with approvals under New Zealand's Animal Welfare Act 1991 approved by the Animal Ethics Committee of the Nelson, Marlborough Institute of Technology (AEC2014-CAW-02).

## Specimen dissection, gut content removal and DNA extraction

The ethanol preserved individuals were placed on a paper towel to remove the excess ethanol, and the frozen individuals were thawed at room temperature until each individual's gut contents could be removed separately. Sterile dissection kits were used for each collection of *M. challengeri* (frozen and ethanol) and the dissection tools were thoroughly cleaned between every individual in each collection; as well as between every individual's hindgut and foregut digesta removal. The gut digesta remained separated throughout the extraction process for each individual, and were placed into separate 1.5 mL microcentrifuge tubes, prior to DNA extraction.

A close microscopic examination of samples of digesta found an almost complete absence of gut items that were sufficiently visually recognisable for them to be reliably assigned to taxa.

Three individuals that contained a large quantity of foregut contents were each mixed independently by vortexing briefly before subsamples of each were taken to be extracted using each of the following DNA extraction kits by following the manufacturers' instructions; Gentra® Puregene® Tissue Kit (Qiagen®, Hilden, Germany), E.Z.N.A.® Mollusc DNA Kit (OMEGA Bio-tek, Inc., Georgia, USA), DNeasy® Tissue and Blood Kit (Qiagen®), and Mo Bio's Powerbiofilm® DNA Isolation Kit (now Qiagen®DNeasy PowerBiofilm Kit). The concentration of recovered DNA was measured for each using the NanoPhotometer® N60 (Implen, Munich, Germany; Table S1). Comparisons of the DNA integrity revealed the Gentra® Puregene® Tissue Kit had superior DNA recovery and was used for subsequent DNA extraction of all *M. challengeri* gut samples.

## Primer selection and PCR protocol

Two universal primer pairs were identified and selected from the literature: mlCOIintF (*Leray et al., 2013*) and jgHCO2198 (*Geller et al., 2013*) targeting a 313 base pair (bp) region of the mitochondrial COI gene, and Uni18SF and Uni18SR (*Zhan et al., 2013*) targeting a 425 bp region of the nuclear 18S rRNA V4 variable region. All primers included Illumina Nextera™ library transposase adapters (Table 1). Primers to block the amplification of host DNA were not used, as previous related work on lobsters (*O'Rorke et al., 2012*; *O'Rorke et al., 2014*) had shown that they also block amplification of related crustaceans, which could be an important part of the diet of *M. challengeri*.

After preliminary trials of a variety of DNA polymerases and buffer mixes, two were tested extensively, Platinum *Taq* (Invitrogen™, Thermo Fisher Scientific Inc., Waltham, MA, USA; Table S2) and the MyTaq™ Red Mix (Bioline, London, UK; Table S3). The PCR temperature profile was optimised to provide the best amplification for both the COI and 18S primers (Fig. S1). Each PCR cycle included a negative control (no DNA added) to check PCR reagents were not contaminated. The PCR products were run on 1.6% agarose

**Table 1 Universal primer pairs used to target the selected regions of the COI and 18S genes.** The COI primers, mlCOIintF and jgHCO2198, target a 313 base pair (bp) region. The 18S rRNA primers, Uni18SF and Uni18SR, target a 425 bp V4 variable region. Illumina Nextera™ library adapters (NexAd) have been added to the primers and are underlined.

| Primer Name | Target | Sequence (adapters underlined) | GC% |
|---|---|---|---|
| mlCOIintF_NexAd (Forward) | COI 313 bp region | 5′ TCG TCG GCA GCG TCA GAT GTG TAT AAG AGA CAG GGW ACW GGW TGA ACW GTW TAY CCY CC 3′ | 48.3 |
| jgHCO2198_NexAd (Reverse) | COI 313 bp region | 5′ GTC TCG TGG GCT CGG AGA TGT GTA TAA GAG ACA G TA IAC YTC IGG RTG ICC RAA RAA YCA 3′ | 42.6 |
| Uni18SF_NexAd (Forward) | 18S V4 425 bp region | 5′ TCG TCG GCA GCG TCA GAT GTG TAT AAG AGA CAG AGG GCA AKY CTG GTG CCA GC 3′ | 54.7 |
| Uni18SR_NexAd (Reverse) | 18S V4 425 bp region | 5′ GTC TCG TGG GCT CGG AGA TGT GTA TAA GAG ACA G GR CGG TAT CTR ATC GYC TT 3′ | 49.1 |

gel and viewed in a Gel Doc™ XR+ (Bio-Rad Laboratories Inc., California, USA). The PCR products were visualised using Gel Red® (Biotium, Fremont, CA, USA).

The effect of both genomic DNA template concentration and BSA concentration were tested to try to reduce PCR inhibition. BSA (1%) was tested with volumes of 1 μl, 2 μl and 5 μl per 25 μl reaction. A range of different DNA dilutions (1:10, 1:50 and 1:100) were also tested.

Inconsistent PCR amplification of both genes was observed in the gut content samples, but not from muscle DNA extracts from the same individuals, likely due to the presence of PCR inhibitors. To test the effects of the presumed PCR inhibitors, gut content template DNA (1μl) was added to tail muscle template DNA (1 μl of 10 ng μl$^{-1}$) in the standard PCRs. Each individual was tested in three replicate PCRs.

## DNA metabarcoding and analyses
### Selection of individuals for metabarcoding
The PCR products for sequencing were selected from six individuals (70.2, 70.3, 70.9, Fro1, Fro2 and Fro3), as they provided consistent and strong amplifications for COI and/or 18S genes. The PCR products encompassed the range of methodological factors that were to be addressed (Table 2). Individual 70.9 was used for comparison of the PCRs to determine which DNA polymerase and buffer mix was better (i.e., samples 1 and 2 versus 3 and 4). Preliminary data suggested the Platinum *Taq* reactions were inferior, so the remainder of the comparisons focussed on the Bioline reactions. The taxa that were identified from the digesta of the foregut and the hindgut were compared for the individuals 70.2 and 70.9 (i.e., samples 1 and 5 versus 2 and 6) to identify if the hindgut digesta could provide useful additional information about the diet, i.e., successive meals may be represented in the hindgut contents. The ethanol and frozen preservation methods were compared using the 18S sequences for three individuals each (i.e., samples 1, 2, 5, 6 and 7 versus 8 and 9), to assess the effects (if any) of differential DNA degradation of the digesta. All individuals were used for the comparison of the database identifications for the COI and 18S genes and thus a preliminary analysis of the diet was evaluated. Potential diet false positives were identified by evaluating the DNA negative.

**Table 2** DNA metabarcoding samples, comprised of six individuals and one DNA negative control (Fro1 and Fro2 are two individuals).

| Sample | Individuals | Digesta source | PCR reagent | Preservation | Gene region |
|---|---|---|---|---|---|
| 1 | 70.9 | Hindgut | Bioline | Ethanol | COI & 18S |
| 2 | 70.9 | Foregut | Bioline | Ethanol | COI & 18S |
| 3 | 70.9 | Hindgut | Platinum *Taq* | Ethanol | COI & 18S |
| 4 | 70.9 | Foregut | Platinum *Taq* | Ethanol | COI & 18S |
| 5 | 70.2 | Hindgut | Bioline | Ethanol | COI & 18S |
| 6 | 70.2 | Foregut | Bioline | Ethanol | COI & 18S |
| 7 | 70.3 | Foregut | Bioline | Ethanol | COI & 18S |
| 8 | Fro1 & Fro 2 | Foregut | Bioline | Frozen | 18S |
| 9 | Fro3 | Foregut and Hindgut | Bioline | Frozen | COI & 18S |
| 10 | DNA Negative | NA | Bioline | NA | COI & 18S |

### DNA purification and pooling of selected samples

NucleoSpin® Gel and PCR Clean-up (Macherey-Nagel, Düren, Germany) was used according to manufacturer's instructions to purify the PCR products and remove fragments of less than 164 bp in length for all selected individuals. The amplified DNA concentration of those products was determined using Qubit™ dsDNA HS Assay Kit (Invitrogen™, Thermo Fisher Scientific Inc.) following the manufacturer's protocol.

The purified COI and 18S PCR products from each sample were pooled and brought to equal molarity where possible. Sequencing was done through New Zealand Genomics Ltd (Auckland, New Zealand) at Massey University (Palmerston North, New Zealand) where indexing occurred using the Nextera™ DNA library Prep Kit (Illumina, San Diego, CA, USA) before sequencing on an Illumina MiSeq™ System ($2 \times 250$ pair-end protocol).

### Metabarcoding protocol

The raw Illumina sequences were analysed through the New Zealand eScience Infrastructure (NeSI; Auckland, New Zealand) high performance computing (HPC) facility. The resulting Illumina metabarcoding sequenced reads were processed by firstly removing the primers (no mismatch tolerated) using fastq-multx (version 1.3.1) and by pairing the reads with SolexaQA++. Low quality 3′ end sequences (Phred scores below 3) were truncated and reads merged using the sequence analysis tool VSEARCH version 2.3.0 (*Rognes et al., 2016*), allowing a maximum of five non-matching nucleotides in the overlap region. Merged reads were quality filtered on the expected error value (<1) and dereplicated. Chimera checking and removal was performed on the dereplicated sequences with the QIIME package (*Caporaso et al., 2010*) based on the *de novo* and reference based methods of Usearch61 (*Edgar, 2010*). Reference based chimera detection of 18S was performed with the largest curated database available (SILVA; *Quast et al., 2013*) and for COI, using the Midori database (*Machida et al., 2017*). The cleaned reads (Table S4 ) were clustered into OTUs with the Swarm methodology (clustering threshold of d2; *Mahé et al., 2014*).

### Sequence assignment and analysis

Representative OTU sequences (seed sequence of each OTU) were taxonomically assigned with the Ribosomal Database Project (RDP) classifier (minimum confidence level of 80%; *Wang et al., 2007*) using the curated databases Protist Ribosomal Reference (PR2; *Guillou et al., 2013*) and SILVA for 18S, and Midori database for COI. Representative sequences were also assigned to reference sequences from the National Center for Biotechnology Information (NCBI) Genbank database (*Benson et al., 2013*), using the megablast option of BLASTn (*Morgulis et al., 2008*) with an e-value threshold of 0.001. Only the best hit for each sequence was kept. NCBI Genbank is an all-inclusive database and thus was used for both the 18S gene sequences and COI gene sequences, however, sequences were only assigned to a taxon if the sequence could be assigned to a genus or species. NCBI Genbank is also not a curated database. The Midori database is specifically for metazoan mitochondrial DNA sequences. The PR2 database consists mainly of nuclear-encoded protistan sequences, and SILVA is an aligned database of small and large subunit rRNA genes. Taxonomically assigned OTUs were filtered for potential PCR or sequencing artefacts by selecting OTUs with five or more sequence reads ("hits"). The resulting filtered OTUs for COI were then further refined to keep only the OTUs that could be identified to either a genus or species level by Midori and the NCBI databases. The same was done for 18S with PR2, SILVA and NCBI. The resulting data for genus or species were used for assessing the methodological factors and preliminary diet which was analysed using Rstudio® (*R Core Team, 2017*) and the collection of R packages in tidyverse (*Wickham, 2017*).

To determine a list of identified taxa that were most likely to form part of the true diet of *M. challengeri*, taxa identified as contamination were removed from the final data. Any sequence reads identified as lobster (Astacidea) were presumed to be host contamination and removed. Sequence reads were also removed from further analysis if they matched to any taxa identified from the sequenced DNA negative sample, or to any taxa that could not be part of the *M. challengeri* diet (e.g., terrestrial species). The final "hit counts" pertaining to digesta are referred to as "diet hit counts".

Taxa were defined as "exclusive" if they were identified in only a single source, such as taxa identified in the foregut digesta but not in the hindgut digesta, or solely identified in one PCR but not another.

## RESULTS

### Factors affecting PCR amplification success

Several important factors were identified that affected PCR amplification success prior to metabarcoding. These factors included PCR inhibition (and its reduction with BSA), optimal template DNA dilution, PCR reagents (particularly the DNA polymerase and buffer), and the success of amplification of different target genes (18S and COI). The success of the PCRs was determined by the presence and intensity of an appropriately-sized fragment on an agarose gel.

The sample DNA concentrations had a wide range (selected individuals ranged from 1.4–290.3 ng $\mu$l$^{-1}$) and, overall, had relatively poor purity ratios (as measured by A260:A280

ratios; Table S5 ). PCR inhibition was minimised (visible increase in DNA amplification on agarose gel) when an optimal volume of 2 μl of 1% BSA was used in each 25 μl PCR, with some beneficial effect when using 1 μl or 5 μl of 1% BSA. PCR inhibition was found to occur inconsistently at a range of DNA concentrations, even with the addition of BSA. The inhibitory effect of the DNA extracted from gut material was demonstrated when amplifying both genes with DNA extracted from tail muscle tissue. PCR amplification from muscle DNA alone was usually strong, whereas PCR amplification was often dramatically reduced when DNA extracted from the gut (digesta DNA) was added. Amplification from digesta DNA was often very inconsistent between PCR replicates. Different dilutions of template DNA (1:10, 1:50 and 1:100) were also tested in an attempt to reduce PCR inhibition and provide optimal DNA amplification at the concentration best suited to the conditions of the individual sample, with the 1:10 dilution proving to result in the strongest DNA amplification overall.

Two different sets of PCR reagents (which differed primarily in their DNA polymerase and buffer) were compared. The Bioline reaction was generally more reliable in DNA amplification, as not only was the intensity of the products generally greater, but the digesta DNA was more likely to produce PCR products for both the COI gene and 18S gene.

When comparing PCR amplification of the two target genes from digesta DNA, in general the COI gene did not amplify as well as the 18S. For several individuals, no PCR products were seen for the COI gene, while for the same DNA, the 18S gene amplified well. In addition, DNA from hindgut contents amplified more readily than DNA extracted from the foregut contents.

## Factors affecting metabarcode sequencing results
### PCR reagents (DNA polymerase and buffer)

A comparison between the Platinum *Taq* and the Bioline PCR reagents was made for individual 70.9, for which both sets of reagents were used (Table 2). Metabarcode sequences from both foregut and hindgut digesta were pooled for each set of reagents (thus comparing samples 1 and 2 with 3 and 4). The total number of taxa identified and the total number of exclusive taxa identified (i.e., those found only in one category) was greater for both the COI and 18S genes when using the Bioline reaction (11 different taxa in total and six exclusive taxa), compared to the Platinum *Taq* reaction (six different taxa identified and one exclusive taxon; Table 3). The total diet hit count (number of sequence reads matching a potential dietary reference sequence in that database—i.e., excluding host matches) was also greater for the Bioline reaction.

Five of the taxa identified by Platinum *Taq* were a subset of those found in Bioline, which had a low summed hit count of 55 compared to Bioline's 468 hit count for the same taxa. Moreover, Bioline identified a higher number of exclusive taxa which averaged a hit count of ~32 hits per taxa whereas Platinum *Taq* only identified one exclusive taxon with a hit count of 13.

**Table 3** **A comparison of the number of assigned diet taxa using the Bioline and the Platinum *Taq* reactions.** The taxa identified in each of the reactions pertain to diet taxa, therefore taxa identified as lobster (Astacidea), matching the DNA negative sample or identified as terrestrial were removed. The 'diet hit count' refers to the number of sequence reads matching a potential dietary reference sequences (identified as diet taxa) in the databases. The 'exclusive taxa' are those taxa found only in one category i.e., in the Bioline reaction but not in the Platinum *Taq* reaction.

|  | Taxa identified in the Bioline reaction (diet hit count) | Taxa identified in the Platinum *Taq* reaction (diet hit count) |
|---|---|---|
| Midori/NCBI (COI) | 9 (648) | 5 (62) |
| NCBI/PR2/SILVA (18S) | 2 (10) | 1 (6) |
| Total taxa | 11 (658) | 6 (68) |
| Total exclusive taxa | 6 (190) | 1 (13) |

**Table 4** **A comparison of the total and exclusive diet taxa identified in the foregut and hindgut digesta.** The taxa identified in each of the reactions pertain to diet taxa, therefore taxa identified as lobster (Astacidea), matching the DNA negative sample or identified as terrestrial were removed. The 'diet hit count' refers to the number of sequence reads matching a potential dietary reference sequences (identified as diet taxa) in the databases. The 'exclusive taxa' are those taxa found only in one category i.e., in the foregut digesta but not in the hindgut digesta.

| Database | Total taxa identified (diet hit count) in the foregut digesta | Total taxa identified (diet hit count) in the hindgut digesta | Exclusive taxa identified in the foregut digesta | Exclusive taxa identified in the hindgut digesta |
|---|---|---|---|---|
| Midori/NCBI (COI) | 3 (448) | 12 (274) | 2 | 11 |
| NCBI/PR2/ SILVA (18S) | 14 (736) | 2 (14) | 13 | 1 |

### Foregut versus hindgut digesta

The taxa identified from the hindgut and foregut digesta of two individuals were compared (i.e., samples 1 and 5 versus 2 and 6). The hindgut digesta contained a higher number of both total taxa and exclusive taxa compared to the foregut digesta, when the sequences were matched to the COI gene databases (Table 4). Conversely, the foregut digesta contained a higher number of total taxa and exclusive taxa with the 18S gene databases (Table 4).

### Sample preservation method

The effects of potential differential DNA degradation due to the method of preservation were more difficult to assess directly, as the same individuals could not be compared across the two conditions, and the comparison was somewhat confounded by being collected in different locations. Instead, broad comparisons were made between the pooled results from the three individuals preserved for each of the ethanol and frozen methods. Firstly, the potential differential effects on preservation of diet DNA was assessed by calculating the diet hit count as a percentage of the overall hit count (for samples 1, 2, 5, 6 and 7 versus 8 and 9; Table 5). No difference was seen between the ethanol and frozen individuals for the 18S results, as both had a diet hit count percentage of 1% (Fro3 was the only frozen individual to have any diet hit counts; Table S6).
**Table 5** **A comparison of the digesta amplification, using the 18S gene region, from ethanol and frozen preserved individuals.** The 'genus/species hit count' refers to the number of sequence reads that matched a high degree of taxonomic resolution (genus and/or species level) from the 18S databases (NCBI, PR2 and SILVA). The 'diet hit count' refers to the number of sequence reads matching potential dietary reference sequences in the databases. The 'diet hit count (%)' is the percentage of the 'diet hit count' out of the total 'genus/species hit count'.

| Preservation | Genus/species hit count | Diet hit count | Diet hit count (%) |
|---|---|---|---|
| Ethanol | 165,566 | 2,028 | 1 |
| Frozen | 94,218 | 902 | 1 |

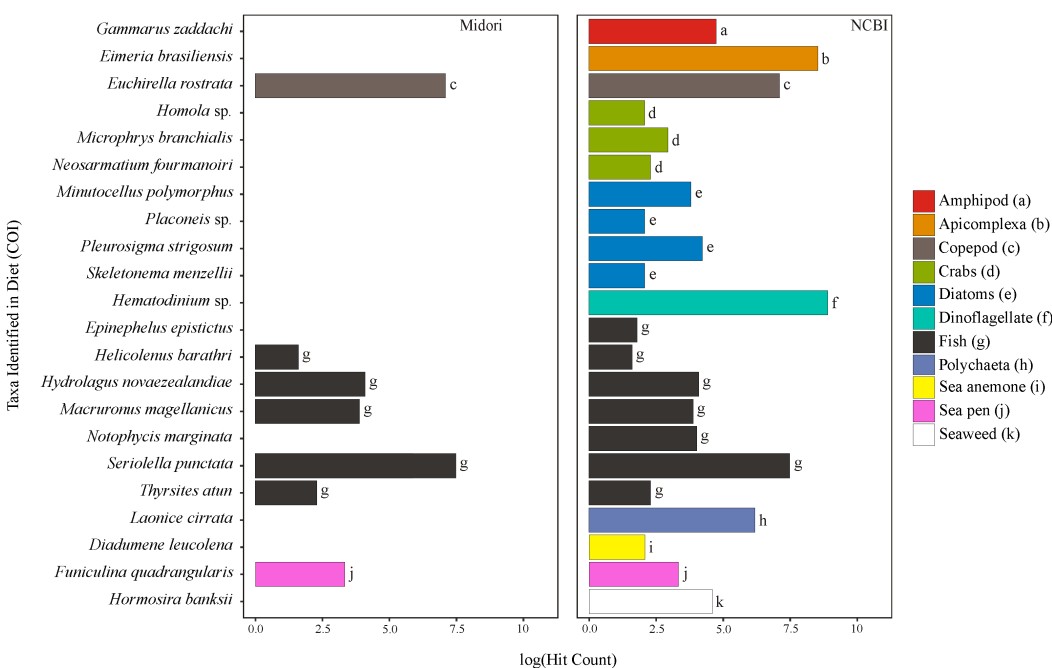

**Figure 1** **Taxa identified from the diet of *M. challengeri* using the Midori and NCBI databases for the COI sequences.**

## Database comparison and preliminary assessment of *M. challengeri* diet

All individuals and their digesta amplified using Bioline were used to provide a preliminary assessment of the diet of *M. challengeri*. A total of 13,541 sequences were used to analyse the diet: 22 taxa were identified from 10,611 COI sequences, and 25 taxa from 2,930 18S sequences. Among the 22 taxa identified from the COI databases were seven fish taxa (*Epinephelus epistictus, Helicolenus barathri, Hydrolagus novaezealandiae, Macruronus magellanicus, Notophycis marginata, Seriolella punctata* and *Thyrsites atun*), crab taxa (*Neosarmatium fourmanoiri, Microphrys branchialis* and *Homola* sp.), a marine worm (*Laonice cirrata*) and an anemone (*Diadumene leucolena*) (Fig. 1; Tables S7 and S8). The 18S databases identified 25 taxa, which included a sea pen (*Umbellula* sp.), salps (*Pyrosoma* sp. and *Ihlea racovitzai*) and many parasites (i.e., apicomplexan *Eimeria variabilis* and
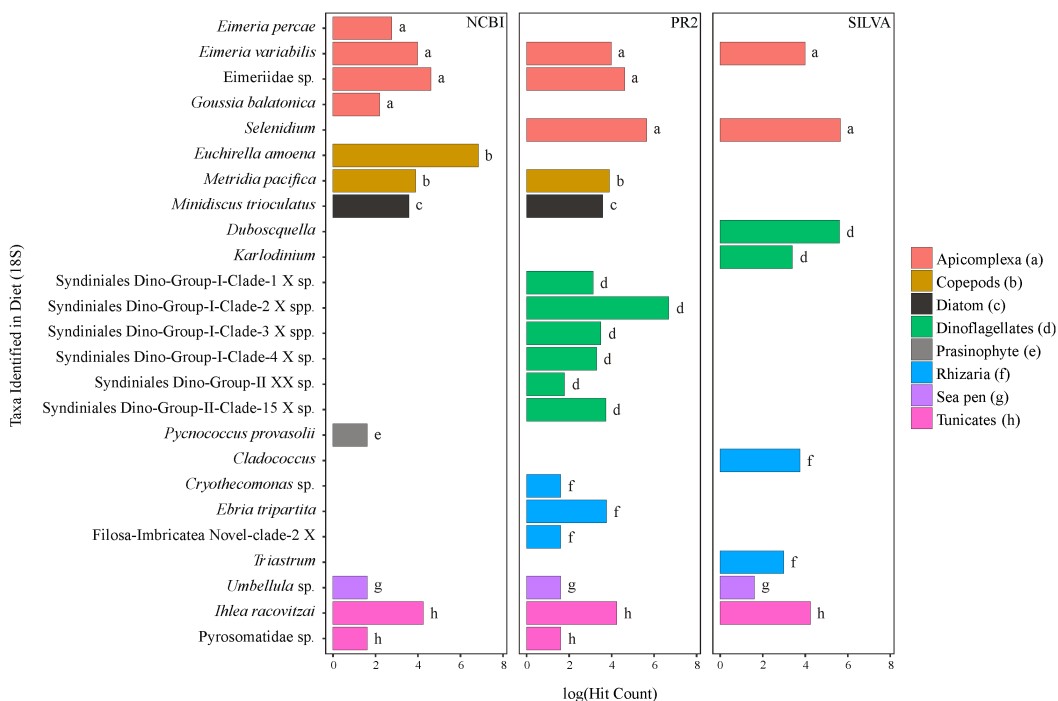

**Figure 2** Taxa identified from the diet of *M. challengeri* using the PR2, SILVA and NCBI databases for the 18S sequences.

dinoflagellate *Duboscquella*) (Fig. 2; Tables S9 and S10). There were no common taxa identified between the reference databases when comparing 18S OTUs to COI OTUs.

# DISCUSSION

## Methodological issues in PCR amplification success
### PCR inhibition

PCR inhibition is a substantial factor in this study. The spectrophotometer ratios of purity provided initial indicators of the presence of inhibitors in the extracted DNA (*Watts, 2014*). An optimal volume of 2 μl 1% BSA successfully reduced PCR inhibition by increasing the amplification efficiency. An additional step effective in reducing PCR inhibition, was 1:10 dilution of the template genomic DNA, thus diluting the PCR inhibitors it presumably contained.

The presence of organic compounds in the foregut (i.e., bile salts in the gastric juices) is thought to be one of the main contributors to PCR inhibition which cannot always be alleviated by BSA (*Lorenz, 2012*; *Schrader et al., 2012*; *Harms-Tuohy, Schizas & Appeldoorn, 2016*). Bile salts often found in vertebrate gastric juices, are similar to an organic compound in crustaceans such as crabs, lobsters and crayfish, and suggests why BSA is not an alleviator in all DNA amplifications tested in this study (*Borgström, 1974*). Different diets would also affect the secretion of the gastric juices, which may play a role in determining which PCR inhibitors are present and in what concentration in different parts of the gut (*Rotllant et al.,*

*2014*). This demonstrates the problematic, high levels of PCR inhibition present in many genomic DNA extracts, and its influence on successfully amplifying gut content DNA. Increased quantitative monitoring of the PCR success, such as by utilising quantitative PCR, may be required in future.

One solution to PCR inhibition may be to pool replicate PCR products together. Another is the addition of the T4 gene 32 protein (gp32), which can be effective against inhibitors for which BSA is not, such as sodium dodecyl sulfate (SDS), sodium chloride and bile salts (*Schrader et al., 2012*).

### PCR reagents (DNA polymerase and buffer)

Different DNA polymerases and buffers are available and are two of the key reagents for DNA amplification, each being susceptible to various components in the biological environment in which they amplify, thus affecting their performance (*Rådström et al., 2004*; *Wolffs et al., 2004*). A substantial difference could be seen visually in the DNA amplification when using the Bioline reaction compared to the Platinum *Taq* reaction. The Platinum *Taq* reaction appeared to be more susceptible to the effects of PCR inhibitors, and the DNA amplification was not as effective as in the Bioline reaction.

### Amplification of the loci

The 18S gene region (∼425 bp), in general, amplified more successfully compared to the COI gene region (∼313 bp) targeted here. It is not common for shorter gene fragments to be more difficult to amplify, but it is relatively common to see some difficulties in amplification of the COI gene in some species, likely due to the greater sequence variation in the primer annealing sites (*Chen, Jiang & Qiao, 2012*; *Lv et al., 2014*). However, when both genes amplified, their PCR product intensity appeared consistent between sample conditions, i.e., if the PCR product was more intense for the foregut digesta for the 18S gene the same would be seen for the COI gene in the same individual.

## Methodological issues in metabarcoding success
### PCR reagents (DNA polymerase and buffer)

Visualization of the PCR amplification can only suggest if amplification is occurring and not necessarily what the quality of amplification is. So although it was visually determined that the Bioline reaction generally resulted in stronger PCR amplifications, it was possible that it was amplifying only a subset of the possible templates compared to that of the Platinum *Taq* reaction. Individual 70.9 amplified equally well for both the Platinum *Taq* and Bioline reactions, and also for both the hindgut and foregut digesta. The DNA sequencing revealed that the Bioline reaction amplified a higher number of taxa from the same individual's gut content DNA.

This was likely due to a higher tolerance of PCR inhibitors from the DNA polymerase and buffer in Bioline. This is suggested by the shared taxa, as Bioline (hit count 468) amplified approximately nine times more diet fragments than Platinum *Taq* (hit count 55). The six exclusive taxa, presumed to have the lower DNA concentrations, identified by Bioline have ∼32 hit counts per exclusive taxon, whereas, Platinum *Taq* only identified

one exclusive taxon with a hit count of 13, thus indicating that the Bioline reaction appears to be more efficient at amplifying DNA from the digesta of *M. challengeri*.

### Foregut versus hindgut digesta

The examination of both the foregut and hindgut digesta provided a broader overview of the diet of *M. challengeri*. There was an increase in the number of taxa when the entire gut digesta was used, compared to using digesta from only the hindgut or the foregut. As ingested food takes a few hours to pass through the foregut and into the hindgut, the examination of the digesta from both may permit the assessment of successive meals over a broader time period before capture, or differences in gut transit times for different dietary items (*Simon & Jeffs, 2008*; *Lee, Hartstein & Jeffs, 2015*; *Kamio et al., 2016*). Also, as the chemical environment (i.e., pH) and degree of degradation of digesta differ between the gut sections, an assessment of both foregut and hindgut digesta appears to maximise the opportunity to detect the widest possible breadth of diet per individual.

It was initially expected that the hindgut contents may provide lower quality sequencing results, because it was more difficult to eliminate host tissue contamination during dissection, the more advanced state of degradation of the food, and the higher ratio of sediment to dietary items. However, a higher number of taxa of smaller marine organisms (i.e., dinoflagellates) were identified in the foregut digesta than in the hindgut digesta, and a higher number of taxa of larger marine organisms (i.e., mostly fish) were identified in the hindgut digesta than in the foregut digesta. Thus, to dismiss sampling from either the foregut or the hindgut digesta would have narrowed the number of taxa that were able to be identified.

### Host contamination

In most metabarcoding studies of diet in predators, host contamination of the PCR amplicons is an issue. However, the overall impact of host contamination can be reduced by introducing several molecular techniques that will help decrease any biased amplification of the host's DNA, and may broaden the taxonomic analysis (*Polz & Cavanaugh, 1998*; *Green & Minz, 2005*; *Vestheim & Jarman, 2008*; *Pompanon et al., 2012*; *Harms-Tuohy, Schizas & Appeldoorn, 2016*; *Devloo-Delva et al., 2018*). PCR "clamping", using additional DNA or peptide nucleic acid (PNA) oligonucleotide primers to bind to, and inhibit PCR amplification of host DNA, is a favoured method for suppressing host DNA amplification in PCRs, and has been successful in diet studies (*Egholm et al., 1993*; *Orum et al., 1993*; *Vestheim & Jarman, 2008*; *Chow et al., 2011*; *O'Rorke et al., 2012*). In this study, PCR clamping was not used, as previous research in lobsters had shown that it also restricted amplification from related crustaceans, many of which could be potential prey of *M. challengeri*. Other diet studies have also shown that the resulting sequences are sufficient to describe the diet without the use of PCR clamping (*Piñol et al., 2014*) and that there would not necessarily be any significant increase in diversity if they were used (*Devloo-Delva et al., 2018*). The metabarcoding results showed that host DNA amplification did not completely outcompete the amplification of prey DNA, and there were sufficient sequence reads to identify a broad range of diet taxa.

### Preservation methods

The quality of DNA extracted from the digesta is affected by the state of degradation due to the digestion rate in the gut (*Rådström et al., 2004*; *Deagle & Tollit, 2007*; *Troedsson et al., 2009*). As the two different preservation methods used would have different rates in which the digesta was preserved, it was thought that one method may provide better preservation of digesta than the other. Determining which preservation method, frozen or ethanol, is better was relatively problematic to assess, due to the relatively high level of inter-individual variation and the confounded nature of the sample collection. Due to logistic constraints on the field sampling, different preservation methods could only be undertaken on different individuals taken from different locations.

The diet hit count percentage from the 18S databases were both ∼1% for ethanol and frozen individuals. Although not a particularly robust test due to sample size, it may suggest that both the frozen and ethanol preservation methods are similarly dependable.

### Amplification of the loci and database review

Although there were difficulties in gaining equal success in the DNA amplification from both genes, together they provided a much broader overview of *M. challengeri* diet at the highest resolution possible.

The 18S gene and the COI gene identified a variety of taxa (Figs. 1 and 2). The COI gene is more commonly used than the 18S gene in studies of molecular taxonomy of large eukaryotes and is one of the pillars of DNA barcoding, making it a sought-after gene region for the discrimination of closely related Metazoan species (*Hebert, Ratnasingham & DeWaard, 2003*; *Derycke et al., 2010*; *Tang et al., 2012*). The 18S gene is more frequently used in molecular taxonomic studies for microbial organisms (i.e., micro-plankton and parasites; *Wu, Xiong & Yu, 2015*).

Using a combination of databases to identify the OTUs, from a selected gene, to genus or species level proved invaluable; where the curated databases (Midori, PR2 and SILVA) could not provide a high resolution identification (i.e., only to order or family), NCBI was usually able to. Also, targeting two genes has allowed for a more inclusive capture of the different taxonomic groups because of the more complete coverage from distinct reference databases. For further analyses of the diet of *M. challengeri* both genes should be used, as this lobster, being relatively small in size, appears to take advantage of the availability of both large and small prey, both dead and alive.

### Preliminary assessment of *M. challengeri* diet

This study undertook the first assessment of *M. challengeri* diet using DNA metabarcoding methods. Although this study investigated only a small number of individuals in this preliminary analysis, it has already provided a fascinating snapshot of their diet, confirming some previous expectations, and providing a broader understanding of their feeding ecology.

The components of the diet were identified at a much finer taxonomic level than any other previous studies undertaken microscopically on any *Metanephrops* species. The diets previously determined microscopically are at a poor taxonomic resolution,

and included crustaceans, fish, annelids and bivalves (*Choi et al., 2008*; *Sahlmann, Chan & Chan, 2011*; *Wahle et al., 2012*). *Metanephrops challengeri* are thought to be benthic foragers and scavengers, relying heavily on chemosensory detection of potential food items (*Major & Jeffs, 2017*). They are likely to be scavengers of fish remains whether it is from trawl debris, sunken carcasses or faeces, as well as foraging for smaller dietary items such as sea pens. Using metabarcoding methods on digesta also allowed identification of parasites that are either from the diet source or from *M. challengeri* individuals sampled. Below, we examine some of the most interesting and prominent taxa identified in the diet of *M. challengeri*, and suggest likely species identifications for those taxa not classified to species or those whose counterparts are more likely to be the closest match.

*Metanephrops challengeri* have been reported to reside at a depth ranging from 140–640 m (*Holthuis, 1991*). Diet taxa were closely matched to several common fish species that reside within this depth range on the Chatham Rise, including *H. novaezealandiae* (ghost shark), *N. marginata* (dwarf codling), *S. punctata* (silver warehou) and *T. atun* (snoek) (*Francis, 1998*; *Ministry for Primary Industries, 2006b*; *Ministry for Primary Industries, 2006a*; *Luna, 2008*; *Ministry for Primary Industries, 2008*; *Priede, 2017*). The OTU identified as *M. magellanicus* is likely instead to be *Macruronus novaezelandiae* (New Zealand hoki), as *M. magellanicus* is located off the southern coast of Chile and Argentina and *M. novazealandiae* is located in the Chatham Rise at depths from 209–904 m (*D'Amato, 2006*; *Connell, Dunn & Forman, 2010*; *Kobayashi, Mizuguchi & Matsuoka, 2014*). *The OTU identified as H. barathri* (sea perch) is likely to be *Helicolenus percoides,* a sea perch (Scorpaenidae) found at depths between 250–700 m *in* the Chatham Rise (*Anderson et al., 1998*; *Horn, Forman & Dunn, 2012*). *Epinephelus epistictus* (grouper) is another OTU that probably belongs instead to its sibling species found in New Zealand waters at the appropriate depth, *Epinephelus octofasciatus*.

The only sea pen to be identified in the COI databases, *Funiculina quadrangularis*, is a common tall deep-sea sea pen that grows on muddy substrates and has been found between 20–2,000 m in New Zealand waters (*Hughes, 1998*). *Umbellula* sp. is a sea pen taxa identified by the 18S databases and has a world-wide distribution in depths 200–6,260 m (*Williams, 1995*; *Williams, Tracey & Mackay, 2014*).

There are several OTUs identified as belonging to the Brachyuran crabs. Many families of this infraorder occur in New Zealand waters in the area of the Chatham Rise (*Wilkens & Ahyong, 2015*), but it is not possible at this stage to identify the OTUs to their likely species from this region. OTUs identified to species in this infraorder include *M. branchialis* (Majidae, spider crabs—12 species found on the Chatham Rise, according to the Ocean Biogeographic Information System (http://www.iobis.org; *OBIS, 2017*), and *N. fourmanoiri* (Grapsidae), although none are known to be found in New Zealand. It is likely that these OTUs belong to genetically similar species that exist at the depth in which *M. challengeri* are found on the Chatham Rise. The *Homola* sp. may be *Homola orientalis*, as this crab is known to be distributed in New Zealand waters, living at depths of 500 m (*Eldredge, 1980*).

The marine polychaete OTU was identified as *L. cirrata*, but this species has not been found in New Zealand waters, and it is likely to be DNA signal from a closely related benthic worm species. Numerous plankton were identified, such as the deep-water salp, *I.*

*racovitzai* (tunicate), which can grow to over 20 mm in diameter and are an appropriate size for *M. challengeri* to handle with their feeding appendages (*Pakhomov et al., 2011*). This species is known to more commonly occur in the high Antarctic cold-water zone, but has been found to occur in waters south of Australia (*Casareto & Nemoto, 1986*; *Pakhomov et al., 2011*; *Ono & Moteki, 2013*). *Ebria tripartita* (Rhizaria) has been reported world-wide, has oval cells 21–35 μm in length and 13–25 μm in width, and is likely to be consumed unintentionally due to their size (*Tong et al., 1998*). It is known to be a grazer of phytoplankton (nanoplanktonic diatoms and dinoflagellates) and has been reported in the Hauraki Gulf, New Zealand (*Gordon et al., 2012*).

It is noticeable that no molluscs were detected, which may be due to more rapid degradation of their soft tissues (*Bell, Tuck & Dobby, 2013*). It will be interesting to examine the ratios of OTU hit counts across a larger number of individuals from several locations.

At present it is impossible to entirely differentiate between OTUs that were directly consumed by *M. challengeri*, and those that were consumed by their prey, i.e., secondary predation. However, it is expected that, due to rapid decomposition, secondary predation is likely to account for only a very small proportion of the taxa detected in *M. challengeri* digesta. It also cannot be dismissed that *M. challengeri* may be cannibalistic, but for the purpose of this study sequences identified as lobsters were removed from the diet data and considered as host contamination.

The parasites that have been identified are likely to be residing either within the gut of *M. challengeri* or within the individuals that were consumed. Parasites known to infect fish were identified, such as *Eimeria percae* which is known to parasitize the European perch, *Perca fluviatilis* (*Molnár et al., 2012*). Syndiniales Dino-Group I and II are known to parasitize a variety of marine organisms, including crustaceans (*Guillou et al., 2010*). Previous studies of gut contents of lobster larvae using DNA methods have also identified a range of microscopic parasites suggesting they may be a common component of the gut (*O'Rorke et al., 2012*; *O'Rorke et al., 2015*).

There was a small proportion of OTU sequences that were not matched to known sequences in the databases and demonstrate the limitations of the reference databases. This is largely due to the small amount of material from New Zealand deep-sea marine life that has been sequenced and uploaded to the various databases. A possible short term solution for the analysis of the diet would be to collect samples of bycatch species from the trawls, identify them, extract their DNA and sequence the genes of interest. This would create a database against which the diet sequences could be compared, potentially revealing more specific matches or even assigning identities to unassigned sequences. Further insight into the diet of *M. challengeri* will come with a greater number of individuals analysed. Data on the availability of prey will be needed in order to determine diet preference, as relative concentration in the diet may simply be related to what is more prevalent in their foraging area.

## CONCLUSION

Many methodological issues have been addressed in this study, which has provided solutions for alleviating them, such as template DNA dilution and BSA addition for minimising PCR
inhibition. The study has also identified considerable variation in PCR inhibition among individual samples, which indicates that it will be necessary to optimise PCR amplification for each sample. The study has also determined the optimum PCR reagents (particularly the DNA polymerase and buffer) and the benefit of using a selection of different databases for assigning OTUs to taxa for different genes.

The preliminary insight from this study into the varied diet of *M. challengeri* provides a foundation for both the production of a nutritional feed for aquaculture and an attractive bait for fishery pots. However, further examination of the diet is clearly needed. Given the variability among individuals, the minimum sample number that is analysed per collection site should be of around 10 individuals that have a moderate amount of digesta for DNA extraction. This will help in quantifying if *M. challengeri* have a diet preference and if it is related to sex, size and/or location. A future study should also proceed in collecting by-catch when *M. challengeri* are trawled, which will assist in determining prey availability, and provide a more complete sequence reference DNA database for comparison. It is clear that there is considerably more insight provided from identification using metabarcoding than from traditional microscopic identification. A further benefit is the store of diet DNA sequences that can be retained for future analysis (against updated reference databases), whereas microscopy results cannot be further analysed. Overall, this study shows there is great promise in analysing *M. challengeri* diet using metabarcoding methods.

## ACKNOWLEDGEMENTS

We would like to thank Dr Ian Tuck (NIWA) and Mr Dean Stotter (NIWA) for providing recorded information and access to samples. We thank the captain and crew of the R.V. Kaharoa in facilitating the collection of the samples for this study. Lastly, thank you to Andrew Stanley (Sanford) and Sophar Rach (Sanford) for assistance with the commercially harvested samples.

### Funding

This study is part of 'Ka Hao te Rangatahi: Revolutionary Potting Technologies and Aquaculture for Scampi' (CAWX1316) funded by the New Zealand Ministry of Business Innovation and Employment. There was no additional external funding received for this study. The funders had no role in study design, data collection and analysis, decision to publish, or preparation of the manuscript.

### Grant Disclosures

The following grant information was disclosed by the authors:
New Zealand Ministry of Business Innovation and Employment.

### Competing Interests

The authors declare there are no competing interests.

## Author Contributions

- Aimee L. van der Reis conceived and designed the experiments, performed the experiments, analyzed the data, contributed reagents/materials/analysis tools, prepared figures and/or tables, authored or reviewed drafts of the paper, approved the final draft.
- Olivier Laroche contributed reagents/materials/analysis tools, authored or reviewed drafts of the paper, approved the final draft.
- Andrew G. Jeffs conceived and designed the experiments, contributed reagents/materials/analysis tools, authored or reviewed drafts of the paper, approved the final draft, oversaw logistics and administration.
- Shane D. Lavery conceived and designed the experiments, contributed reagents/materials/analysis tools, authored or reviewed drafts of the paper, approved the final draft.

## Ethics

The following information was supplied relating to ethical approvals (i.e., approving body and any reference numbers):

Specimens for this study were collected in accordance with approvals under New Zealand's Animal Welfare Act 1991. The transport and holding of the scampi, as well as the experimental procedures, were approved by the Animal Ethics Committee of the Nelson Marlborough Institute of Technology (AEC2014-CAW-02).

## Field Study Permissions

The following information was supplied relating to field study approvals (i.e., approving body and any reference numbers):

A special permit (#549) for *M. challengeri* collection was provided by New Zealand's Ministry of Primary Industries.

## DNA Deposition

The following information was supplied regarding the deposition of DNA sequences:

The sequences of the scampi diet matching genus/species are available as Supplemental Files; Table S8 contains the COI sequences and Table S10 contains the 18S sequences.

## Data Availability

Figshare: https://doi.org/10.17608/k6.auckland.5928400.

## Supplemental Information

Supplemental information for this article can be found online at http://dx.doi.org/10.7717/peerj.5641#supplemental-information.

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
