# Peer review of "Preliminary analysis of New Zealand scampi (Metanephrops challengeri) diet using metabarcoding"

_PeerJ, doi:10.7717/peerj.5641_

## Round 0.1 · original submission · Major Revisions

The reviewers have identified several major problems regarding both the overall study design and the choice of particular experimental methods. These issues need to be addressed in the revised manuscript.

Reviewer 1 ·

Basic reporting

In general, the manuscript is well-written, the research problem is clearly defined and the sufficient context is provided.

Experimental design

The research question is clearly defined and meaningful. However I feel that the authors could have done much more for the exploration of this question (one may argue that this remark could be applied to any study but I have several specific points to consider).
First, the authors used only two ways to get rid of inhibitors - the dilution and the addition of BSA. Both have limited value - the dilution can be not efficient in case of low DNA content and/or high inhibitor content and BSA is not efficient for several types of inhibitors. PCR inhibitors are well known issue for many metagenomic samples (soil, food, fecal samples). There are many protocols and commercially available kits for the removal of these inhibitors, they are widely used in metabarcoding studies (see, e.g. https://www.frontiersin.org/articles/10.3389/fmars.2016.00096/full). Why they were not used?
Second, the DNA samples obviously contain the host DNA which is coamplified together with the DNA from the digested material. Moreover, the host DNA will amplify with much higher efficiency because it is less degraded. Thus it can mask the presence of certain taxa in teh digested material, especially when samples are sequenced with low coverage (by the way, the number of reads per sample in not mentioned anywhere in the manuscript; it is an important information and should be reported). To overcome this, the blocking oligonucleotides specific to the host DNA are used (see e.g. https://www.ncbi.nlm.nih.gov/pmc/articles/PMC5653352/, https://www.ncbi.nlm.nih.gov/pmc/articles/PMC5112926/). This study would greatly benefit from this approach, given that the diet hit count very low.

Validity of the findings

The manuscript has several methodological shortcomings that decrease the validity and the usefulness of the findings.

Additional comments

- please consider the factor of GC content as it greatly affects PCR efficiency.
- there are several typos and inconsistencies, for example at lane 534: "has oval cells 21-35 pm". What is pm? Picometers? I doubt that the cells are so small.

Reviewer 2 ·

Basic reporting

A generally well-written manuscript with sufficient background information, relevant literature, and good structure. A few suggestions for improvement:
line 41: Regarding the Balss 1914 comment on high value of deep sea lobsters. Reference not listed in literature cited.
Some of the methodological details could be moved to a supplemental materials section to make the Methods less involved and lengthy, thus more readable. For example, (Line 144): some of the details in the paragraph describing the PCR cycle.
Discussion is quite long relative to the rest of the paper.

Figure 1: Not necessary as it is not very informative. No labeling of islands or depth strata. A figure like this isn't necessary to show one sampling location.

Experimental design

lines 109-111: "For comparison, frozen individuals of commercially harvested NZ scampi were supplied by Sanford Ltd 110 (Auckland, New Zealand), their location of harvesting unknown." This is a problematic confounding factor for comparison - different locations, different diets? Better to have frozen some of their own samples in manner similar to commercial enterprises.

Also missed a big opportunity to compare these methods to the standard gut content analysis. Could have identified the stomach contents microscopically, then conducted these genomic analyses

A laboratory-based food preference & gut content study would also be a better way to conduct the initial analysis to cross-check the methods before trying to apply them to a field data set.

Validity of the findings

Validity of results suffers from low sample sizes and are confounded by all the distinctions.

lines 280-285: The contradictory results for the fore- and hind-gut comparisons of the two methods is perplexing and suggests problems with the resolution of the two methods. Which is correct? I agree with authors conclusions in the Discussion that using the entire gut contents seems the best approach.

lines 307-310: "The databases (18S verse COI) did not confidently identify any of the same organisms and the taxonomic resolution and level of confidence varied for the same OTU between databases when assessing the same gene (Table S1 and Table S2)." This result does not instill much confidence.
Line 228: any sequence reads identified as lobster presumed as contamination but some lobsters will eat other lobsters so you would discount this food source?

Additional comments

A generally well-prepared manuscript, but the results are not clear-cut and raise more questions than provide solutions - which is not what one desires in a paper focusing on methods development. The experimental design also does not lend itself particularly well to the over-arching goals of the study. Specifically, the study comes across as a latent attempt to do something more sophisticated with basic trawl data, in this case to use genomic metabarcoding of stomach contents of a deep sea scampi to try to ascertain its diet. However, the authors did not compare their results to a robust, comparable control data set (e.g., their commercial frozen samples), they didn't take the time to conduct a standard gut content study for comparison, and they did not attempt a laboratory food preference study to confirm the methodology. These are lethal weaknesses.

Reviewer 3 ·

Basic reporting

On the up side, this study was an interesting use of the metabarcoding approach and provides new information about a species that had very little prior data. That could be a reasonable justification for publishing the results. On the down side, it only reported data from a very small sample size, which makes it difficult to determine whether the results were several points in a “noisy” system and represented an accurate sample of the whole system. The use of BSA to suppress PCR inhibitors and a comparison of preservation methods was not novel, but it was helpful information to report for such a study. Usually this information is briefly reported in the results section and does not become a major component of the discussion. The comparison of polymerase enzymes was limited to two manufacturers and, because suppliers often switch and adjust their products, this may provide limited or short term value. The fact that this study was the first to report information about the diet of Metanephrops challengeri (six samples from one location) is the important result that will provide some benefit to the science community. However, I am concerned that the small sample size might be misrepresenting the total amount of diversity in the diet and the best methodologies to use. One way to address this concern would be to provide a justification that the diet variability is expected to be narrow (among individuals and sites, and over time e.g. seasonal differences) and so a small sample is all that is needed to provide an accurate picture. Alternatively, if the authors believe that their study is not about the species that were detected in the diet, then they should discuss the shortcomings of the sample size and what sample design might be best for this, and similar, species in a full scale study.

• The abstract did not contain any information about the DNA sequencing results (e.g. the number of sequences returned and searched)
• L28: “resulted in better results” seems awkward.
• L65: “through time” is odd, change to "among phylogenetically distinct taxa”
• L66: “powerful” is an ambiguous and loose term that is used to describe the metabarcoding method
• L70” “high taxonomic resolution” - does this mean a wide range of species can be detected, or that the detection can be made to the species (e.g. fine scale) level?
• L73: Sequencing costs are a direct cost, but if all indirect costs are considered (e.g. people’s time) then the new method might be similar to the old method. The overall science system is set up to easily measure direct costs (and largely ignore indirect costs). Presumably it is hard to measure the real costs of the two methods.
• L83-85: A trial of only two polymerases seems small. It must be hard to work out what is a good sample of polymerases. How many are available on the market? If manufacturers/suppliers change over time the results will be obsolete quite soon.
• L93: It would be better to use the genus-species name instead of the common name
• L126: The DNA concentration of the samples was measured but never reported. Was it used to inform the dilutions for the comparisons of samples? The lack of standardisation of the DNA template concentration and the small sample size means there is a good chance that the results are somewhat uninformative.
• L140: An amount of 1 µl of DNA was added, but would it be better to be consistent with the way all of the other reagents are list and include a concentration of DNA that was added.
• L324: What about the low DNA concentration? Add in something here about how low template concentration might be an important factor for PCR success.
• The labels on Table 1 seem to show that only five different samples were used. Sample 8 looked like in was a combination of Fro1 and Fro2.
• The term "PCR reactions” is repeating the word ‘reactions’, which is a bit strange.

Experimental design

Validity of the findings

---

## Round 0.2 · Minor Revisions

As noted, there still remains a number of technical issues, the most important one being that of the low number of characterized reads.

Reviewer 1 ·

Basic reporting

no comment

Experimental design

Please provide the table on the results of DNA extraction with alternative kits. By the way, did you analyzed the DNA integrity (using capillary or agarose gel electrophoresis)?

Regarding GC content, although GC content of primers is also important, I meant the content of the templates. High GC templates are amplified less efficiently than those with low or moderate GC. Please provide GC content values for M. challengeri marker sequences and for those of diet sequences.

Thank you for providing read counts, this information is important though raises additional questions. Why the number of reads for COI in Fro1 & Fro2 is so low? The number of COI paired reads for 70.9 hindgut sample amplified with Platinum taq is high but the number of merged reads is two orders of magnitude lower. Why? Where the remaining reads come from?

Validity of the findings

no comment

Additional comments

no comment

---

## Round 0.3 · accepted · Accept

The remaining concerns have been addressed.

#